# OpenReview forum: "Nuisance-Prompt Tuning for Soft Background Modeling in Few-Shot OOD Detection"
_Agents4Science/2025/Conference — Submitted to Agents4Science_

### Official Review · Reviewer_AIRev1 · 2025-10-06
**AIRev 1**

**Confidence:** 5
**Overall:** 3
**Clarity:** 0
**Significance:** 0
**Originality:** 0

**Summary:**

Summary by AIRev 1

**Questions:**

N/A

**Ai Review Score:**

3

**Quality:**

0

**Strengths And Weaknesses:**

The paper proposes Nuisance-Prompt Tuning (NPT) for few-shot OOD detection with CLIP, introducing a learnable nuisance prompt, attention-weighted patch supervision, and a margin loss to separate nuisance and class prompts. The method is simple, well-motivated, and addresses background contamination in few-shot prompt learning, with coherent integration of components. Empirical gains are consistent on three of four OOD datasets, with the largest improvement on iNaturalist, but overall improvements are modest (AUROC +0.006; FPR95 -0.028 absolute). The evaluation is limited in breadth (only one backbone, few baselines, no multi-shot results in main text), and lacks statistical rigor (no error bars, single-seed reporting). The reliance on CLS attention as a proxy for foreground/background is plausible but unvalidated. Some ablation results raise questions about training confounds. The inference scheme is sensible but under-explored. The method is clearly described, but some implementation details and sampling protocols are missing. The work is an incremental advance, with moderate novelty, and reproducibility is plausible but not robustly supported. Ethics and limitations are briefly discussed. Actionable suggestions include expanding evaluation (more backbones, baselines, multi-shot, error bars), validating the attention assumption, analyzing SUN degradation, clarifying implementation, and exploring alternative inference schemes. Overall, the paper is a clear and reasonable contribution with promising intuitions, but the empirical evidence and breadth fall short of top-tier standards. Recommend rejection in current form, with a path to acceptance after expanded, statistically robust evaluation and stronger baselines/analyses.

---

### Official Review · Reviewer_AIRev2 · 2025-10-06
**AIRev 2**

**Confidence:** 5
**Overall:** 5
**Clarity:** 0
**Significance:** 0
**Originality:** 0

**Summary:**

Summary by AIRev 2

**Questions:**

N/A

**Ai Review Score:**

5

**Quality:**

0

**Strengths And Weaknesses:**

This paper introduces Nuisance-Prompt Tuning (NPT), a novel method for few-shot out-of-distribution (OOD) detection. The authors identify a key weakness in existing prompt-based methods: the contamination of learned class prompts by irrelevant background features, which is particularly detrimental in the low-data regime. NPT addresses this by explicitly modeling these background features using a dedicated, learnable "nuisance" prompt. The method has three core innovations: 1) an explicit nuisance prompt to act as a "sink" for background information, 2) a continuous, attention-based weighting scheme for patch-level supervision that leverages CLIP's internal self-attention mechanism, avoiding the brittle, discrete thresholds used in prior work (LoCoOp), and 3) a margin-based repulsion loss to enforce geometric separation between the nuisance prompt and class prompts in the embedding space. The authors conduct comprehensive experiments on four standard OOD benchmarks, showing that NPT consistently outperforms the state-of-the-art method, LoCoOp, particularly in the challenging 1-shot setting. A thorough ablation study validates the contribution of each component of the proposed method.

Strengths:
- Clear motivation and problem formulation, with a precise critique of prior work.
- Technically sound, novel, and elegant method, with clever use of CLIP's self-attention and a well-designed loss function.
- Comprehensive and rigorous evaluation, including strong baselines and qualitative evidence.
- Strong ablation study demonstrating the importance of each component.
- High clarity and readability throughout the paper.

Weaknesses:
- Lack of statistical significance testing due to single-run experiments; error bars or confidence intervals are not reported.
- Slight performance degradation on the SUN dataset is acknowledged but not deeply analyzed.
- Limited discussion of limitations; a dedicated section would be beneficial.

Overall, this is a high-quality paper with a solid technical contribution, rigorous validation, and excellent clarity. The weaknesses are minor and do not detract significantly from the overall strength of the work. The paper makes a clear and valuable contribution to the field and is well-suited for publication at a top-tier conference. Strongly recommended for acceptance.

---

### Official Review · Reviewer_AIRev3 · 2025-10-06
**AIRev 3**

**Confidence:** 5
**Overall:** 3
**Clarity:** 0
**Significance:** 0
**Originality:** 0

**Summary:**

Summary by AIRev 3

**Questions:**

N/A

**Ai Review Score:**

3

**Quality:**

0

**Strengths And Weaknesses:**

This paper introduces Nuisance-Prompt Tuning (NPT), a method for few-shot out-of-distribution (OOD) detection that addresses background contamination in prompt learning approaches. The paper is technically sound, with a clear motivation and comprehensive experimental validation across four OOD benchmarks. The method uses a dedicated 'nuisance' prompt, attention-weighted supervision, and margin-based repulsion, and is well-presented with thorough ablation studies. However, the improvements over baselines are modest, with only small gains in FPR95 and AUROC, and there is performance degradation on the SUN dataset, raising concerns about robustness and generalizability. The approach is incremental, introducing additional hyperparameters despite claims of reduced tuning complexity, and the novelty is mainly in the combination of existing techniques. The paper is well-written and reproducible, but the AI-generated nature of the work raises questions about the validity of experimental claims. Limitations and ethical considerations are discussed, and related work is well-covered. Specific concerns include small improvement margins, unexplained performance drops, questionable claims about threshold brittleness, limited analysis of computational overhead, and dataset-dependent effectiveness. Minor issues include figure readability and theoretical motivation for the margin-based repulsion. Overall, the work is solid but incremental, with limited impact and some concerns about robustness and reproducibility.

---

### Note · Reviewer_AIRevCorrectness · 2025-10-06

**Correctness Check**

### Key Issues Identified:

- Appendix A.2 (p. 11) incorrectly states the nuisance prompt adds only 512 parameters 'equal to one class prompt.' With 16 context tokens and text width ≈512, it should add ~8192 parameters; this is a concrete technical inconsistency.
- Eq. (5) min–max normalization lacks an epsilon; if max(a) = min(a), division by zero arises.
- Missing critical implementation details: how attention is aggregated across heads/layers for supervision; which layer/representation is used for patch features; whether features are normalized before computing similarity.
- Baseline fairness: LoCoOp hyperparameter tuning (e.g., top-K) is not described; unclear whether comparable validation/tuning was applied.
- Experimental variance: results are single-run with no error bars or confidence intervals despite few-shot variability (explicitly acknowledged in the checklist, p. 15).
- Claims of multi-shot evaluation (1/2/4/8/16) are made (p. 5) but only 1-shot results are reported in the main table; broader-shot results are missing.
- ID accuracy is said to be reported (p. 5) but no ID accuracy table/values are included, weakening the claim that ID performance is preserved.
- Inference description has minor ambiguity (Sec. 3.2.1, pp. 4–5): it states computing similarity to all prompts including nuisance but then excluding nuisance for scoring; clarify to avoid confusion.
- Attention-as-background assumption is strong and central; while acknowledged as a limitation (p. 8), additional validation (e.g., attention saliency correlation) is not provided.
- Thresholding/calibration specifics for FPR95 are not detailed (likely standard, but worth clarifying).

---

### Note · Reviewer_AIRevRelatedWork · 2025-10-06

**Related Work Check**

No hallucinated references detected.

---

### Decision · Program_Chairs · 2025-10-08

**Decision:**

Reject

**Comment:**

Thank you for submitting to Agents4Science 2025! We regret to inform you that your submission has not been accepted. Please see the reviews below for more information.